# Hypoxia-Regulated lncRNA USP2-AS1 Drives Head and Neck Squamous Cell Carcinoma Progression

**DOI:** 10.3390/cells11213407

**Published:** 2022-10-28

**Authors:** Jianmin Tang, Zheng Wu, Xiaohang Wang, Yanli Hou, Yongrui Bai, Ye Tian

**Affiliations:** 1Department of Radiotherapy and Oncology, The Second Affiliated Hospital of Soochow University, Suzhou 215004, China; 2Institute of Radiotherapy and Oncology, Soochow University, Suzhou 215004, China; 3Department of Radiation Oncology, Ren Ji Hospital, Shanghai Jiao Tong University School of Medicine, No.160 Pujian Road, Shanghai 200127, China

**Keywords:** long non-coding RNA, USP2-AS1, cell proliferation and growth, DCAF13 E3 ligase, head and neck squamous cell carcinoma

## Abstract

The role of hypoxia-regulated long non-coding RNA (lncRNA) in the development of head and neck squamous cell carcinoma (HNSCC) remains to be elucidated. In the current study, we initially screened hypoxia-regulated lncRNA in HNSCC cells by RNA-seq, before focusing on the rarely annotated lncRNA USP2 antisense RNA 1 (USP2-AS1). We determined that USP2-AS1 is a direct target of HIF1α and is remarkably elevated in HNSCC compared with matched normal tissues. Patients with a higher level of USP2-AS1 suffered a poor prognosis. Next, loss- and gain-of-function assays revealed that USP2-AS1 promoted cell proliferation and invasion in vitro and in vivo. Mechanically, RNA pulldown and LC–MS/MS demonstrated that the E3 ligase DDB1- and CUL4-associated factor 13 (DCAF13) is one of the binding partners to USP2-AS1 in HNSCC cells. In addition, we assumed that USP2-AS1 regulates the activity of DCAF13 by targeting its substrate ATR. Moreover, the knockdown of DCAF13 restored the elevated cell proliferation and growth levels achieved by USP2-AS1 overexpression. Altogether, we found that lncRNA USP2-AS1 functions as a HIF1α-regulated oncogenic lncRNA and promotes HNSCC cell proliferation and growth by interacting and modulating the activity of DCAF13.

## 1. Introduction

Head and neck squamous cell carcinoma (HNSCC) comprises a heterogeneous group of tumors that arise from the squamous epithelium of the oral cavity, oropharynx, larynx, and hypopharynx [1,2]. Most HNSCC cases are diagnosed at advanced stages due to the lack of early symptoms and early detection, which contributes to poor five-year survival rates of 2–9% [3]. There are many unknowns in the field of detailed HNSCC molecular pathogenesis. The tumorigenesis and progression of HNSCC result from a series of gene mutations. Previous studies revealed that the mutations of KRAS, P16, CDNK27, P53, and SMAD4 contribute to the development of HNSCC [4].

Hypoxia is a typical tumor microenvironment, especially in solid tumors, including HNSCC [5]. Hypoxia-inducible transcription factors (HIFs), mainly including HIF1α, HIF2α, and HIF1β, are the key transcription factors and regulators of the cellular response to hypoxia [6]. Under hypoxic conditions, these HIFα subunits, especially HIF1α, are stabilized and translocated to the nucleus to form heterodimers with HIF1β, which are then specifically bound to the promoter regions of HIF target genes, thereby inducing gene transcription [7]. Proteins encoded by HIF-1 target genes such as PDK1, LDHA, and GLUT1 are involved in glucose and energy metabolism [8], and many other targets are associated with multiple aspects of tumorigenesis and progression [9].

The investigation of long non-coding RNAs (lncRNAs) has recently gained widespread attention as a new route for cancer development regulation [10]. It has been shown that lncRNAs function primarily through their interaction partners such as chromatin DNA, proteins, and RNAs [11,12]. Many oncogenic and tumor-suppressive lncRNAs have been reported to regulate tumor cell proliferation and growth, metabolism, and metastasis [13]. These lncRNAs have been regarded as potential biomarkers and therapeutic targets for cancer diagnosis and treatment [14]. Hypoxia-regulated lncRNAs have also been uncovered in cancer initiation and progression in the past decade [15].

USP2-AS1 is a poorly studied long non-coding RNA. Through a comprehensive screening of hypoxia-regulated lncRNAs in HNSCC cells, we found that USP2-AS1 is a hypoxia-responsive lncRNA that is directly regulated by HIF1α. Further investigation revealed that USP2-AS1 was upregulated in HNSCC and could be used to predict poor prognosis. A biological and mechanical investigation revealed that USP2-AS1 promotes HNSCC cell proliferation and invasion by modulating DCAF13 signaling.

## 2. Materials and Methods

### 2.1. Cell Lines

HEK-293T and the HNSCC cell lines CAL27 and FaDu were purchased from ATCC (Manassas, VA, USA). All these cell lines were maintained in DMEM medium with high glucose and sodium pyruvate; 10% FBS (fetal bovine serum, sigma); 100 units/mL penicillin; and 100 µg/mL streptomycin (GIBCO). The authentication of the cells was performed by short tandem repeat analysis every year. Cells were monitored for mycoplasma contamination using the PCR methods [16] every 3 months. For hypoxia incubation, cells were cultured under 1% O_2_, 5% CO_2_, and 37 °C.

### 2.2. Lentivirus Infection and Establishment of Stable Cell Lines

All the plasmids including pLVX-USP2-AS1 (USP2-AS1); pLVX-DCAF13-Flag; pLVX vector (EV); shRNAs targeting USP2-AS1 (sh#1, sh#2, sh#3); shDCAF13; and the negative control (shNC) were purchased from Shanghai Zhuyi biotech (Shanghai, China). The lentiviruses were grown in HEK-293T cells and collected from the supernatant 24 and 48 h after transfection. These lentiviruses were transduced FaDu or CAL27 cells, and cells were selected using 2 μg/mL puromycin (Millipore, Boston, MA, USA) after 48 h infection.

### 2.3. RNA Isolation and Quantitative Real-Time PCR (qRT-PCR)

Total RNAs were isolated from indicated HNSCC cells using TRIzol reagent (Invitrogen, Carlsbad, CA, USA) according to the manufacturer’s instructions. Two micrograms of total RNA were treated with DNase I at 37 °C for 15 min and reverse transcribed using the MMLV system (Promega, Madison, WI, USA) for qRT-PCR tests. The qRT-PCR assays were performed using an ABI 7900 RT-PCR system with SYBR Green Real-time PCR Master Mix (ABI, Waltham, MA, USA). *ACTB* mRNA was used as a reference control. The sequences of primers are listed in Appendix A.

### 2.4. Western Blot

Indicated cells were seeded into 6 cm plates at 4 × 10^5^ cells per plate for 48 h. Cell lysates were then subjected to a standard western blot procedure [17]. These membranes were all incubated with primary antibodies overnight at 4 °C. The next day, these membranes were washed with PBS 3 times and then incubated with anti-mouse (Abcam, ab6728) or rabbit IgG-HRP (Abcam, ab6721). The primary antibodies were as follows: DCAF13 (Abcam, ab195121); lamin B (Abcam, ab32535); β-tubulin (Proteintech, 10094-1-AP); P53 (Abcam, ab26); ATR (Abcam, ab241067); HIF1α (Novus, NB100-105); HIF1β (Novus, NB100-124); and β-actin (Santa Cruz Biotechnology, sc69879). All the primary antibodies for western blot assays were diluted at 1:1000 in 2% BSA (*m*/*v* in PBS).

### 2.5. CCK8 Cell Viability Assays

We seeded 10^3^ indicated cells into 96-well plates and then cultured them under normoxia (21% O_2_) or hypoxia (1% O_2_); the absorption (OD450) was measured with a CCK-8 kit (Dojindo Laboratories, Kumamoto, Japan) according to the manufacturer’s instruction every 24 h.

### 2.6. Cell Colony Formation Assays

The indicated cells (500 cells/3 mL) were seeded into 12-well plates and incubated for 8–12 days at 21% O_2_ (normoxia), 37 °C, and 5% CO_2_, and the media were changed every three days. The cell colonies were washed once with PBS and then fixed with 0.5% paraformaldehyde for 20 min. Subsequently, the colonies were stained with crystal violet and then counted. All the experiments were conducted three times with similar results.

### 2.7. Cell Cycle Analysis

Cell cycle analysis was performed according to a previous study [18]. Briefly, indicated cells were washed with PBS, trypsinized and collected, and then fixed with 70% pre-coated ethanol at −20 °C until examination. Cell samples were treated with 20 mg/mL RNase (Sigma-Aldrich) for 1 h at 37 °C; labeled with 20 mg/mL propidium iodide (PI, Sigma-Aldrich); and then assessed by FACS Calibur flow cytometry (BD).

### 2.8. Cell Invasion Assay

Indicated stably transfected FaDu and CAL27 cells were trypsinized and re-suspended in serum-free DMEM. Approximately 5 × 10^4^ cells/0.1 mL, pre-coated with matrigel (BD) and seated on a 24-well plate, were placed into the upper chamber (Millipore). DMEM containing 5 % (*v*/*v*) FBS was added to the bottom chamber. Cells were incubated at 37 °C under normoxia or hypoxia and allowed to invade through for 48 h. After 48 h, non-invading cells were scrapped using a cotton swab, and the chambers were then fixed and stained with 0.1 % (*w*/*v*) crystal violet. The invaded cells were then counted under a microscope at a magnification of 200×. At least five grids per filter were counted, and the experiments were repeated three times.

### 2.9. Luciferase Reporter Assay

The USP2-AS1 promoter region (−3000 bp~+174 bp) containing all 11 putative hypoxia-response elements (pHREs1+2) or truncations (pHREs1 (−3000 bp~−1700 bp) and pHRES2 (−1701 bp~+174 bp)) was constructed in pGL3-based vectors. To determine the effect of HIF1α on the USP2-AS1 promoter, these pGL3-based constructs plus Renilla luciferase reporter plasmids were individually transfected into HEK-293T cells. After culturing under normoxia or hypoxia for 24 h post-transfection, firefly and Renilla luciferase activity were measured by a dual-luciferase reporter assay system (Promega). The ratio of firefly luciferase to Renilla activity was calculated for each sample.

### 2.10. Chromatin Immunoprecipitation (ChIP) Assay

ChIP assays were performed according to a previous report [19]. Briefly, a ChIP assay kit (Beyotime) was used, and FaDu cells (pre-incubated under hypoxia for 12 h) were crosslinked with formaldehyde and sonicated to an average size of 200 to 500 base pairs. Cell lysates were precleared with protein A/G beads before incubation with protein A/G beads coated with the anti-HIF-1α antibody (Novus). Anti-mouse IgG was used as a negative control. Crosslinked DNA released from the protein–DNA complex was purified by a DNA Extraction Kit (Beyotime), and the eluted DNA was further subjected to qRT-PCR. The specific primers used for ChIP-qPCR are presented in Appendix A.

### 2.11. RNA Pulldown Assays and LC–MS/MS

The biotin-labeled RNA pulldown assays were performed as described previously [18]. Briefly, USP2-AS1 and control (USP2-AS1-AS) were transcribed in vitro; supplemented with U-biotin using an RNA transcription kit (#P1320, Promega) from linearized constructs; and purified with TRIzol reagent (Invitrogen). In vitro transcripted RNAs were incubated in RNA structure buffer (10 mM Tris-HCl, pH = 7; 0.1 M KCl; and 10 mM MgCl_2_) and then heated to 72 °C for 2 min to form the proper secondary structures. These RNAs were then incubated with FaDu cell lysates (10^8^ FaDu cells cultured under hypoxia for 12 hours) at 4 °C for 4 h, followed by incubation with streptavidin beads (Invitrogen) at room temperature for 1 h. After 5 washes, the pulldown complexes were analyzed by LC–MS/MS identification or western blotting.

### 2.12. RNA Immunoprecipitation (RIP) Assays

RIP assays were performed as described previously [20]. Briefly, 2 × 10^7^ FaDu cells with an overexpression of Flag-tagged DCAF13 were crosslinked with 0.3% formaldehyde in medium for 10 min at room temperature, followed by neutralization with 250 mM glycine for 5 min (the FaDu cells were cultured under hypoxia for 12 h). After washing with cold PBS, the cells were lysed in RIPA buffer (50 mM Tris, pH 7.4; 150 mM NaCl; 1 mM EDTA; 0.1% SDS; 1% NP-40; 0.5% sodium deoxycholate; 0.5mM dithiothreitol; RNase inhibitor; and protease inhibitor cocktail), followed by sonication on ice and DNase Ⅰ treatment. The cell lysates were pre-cleared with protein A/G beads for 30 min and then incubated with Flag antibody (BD bioscience) or equivalent mouse IgG (GeneTex) at 4 °C overnight. On the second day, the antibody was precipitated by incubation with protein A/G beads. After 5 × 10 min washes, the RNA samples were extracted with Trizol reagent (Invitrogen) and detected by qRT–PCR. These protein samples were subjected to western blots.

### 2.13. Subcutaneous Tumor Formation Assays

We subcutaneously injected 5 × 10^6^ indicated stable cells into both flanks of 5 six-week-old BALB/c nude mice (Linchang biotech). Mice were sacrificed after 24 days, and the tumors were photographed and weighed. All animal experimental procedures were conducted strictly in accordance with the National Institutes of Health guidelines for the care and use of laboratory animals (NIH Publication No. 8023, revised 1978) and approved by the committee for the humane treatment of animals at Shanghai Jiao Tong University.

### 2.14. Subcellular Fractionation

The isolation of the cell nucleus and cytoplasm in FaDu cells was performed using NE-PER™ Nuclear and Cytoplasmic Extraction Reagents (Thermo), and the procedure was conducted according to the manufacturer’s instructions. Briefly, 1 × 10^5^ FaDu cells were divided into two parts; one part was treated with Trizol for total RNA extraction, and the other part was used for subcellular fractionation. For the latter procedure, the cell pellet was suspended in 100 μL CERⅠ, incubated on ice for 10 min, added to 5.5 μL CERⅡ, and then centrifuged at 16,000 rpm for 10 min at 4 °C. The supernatant (cytoplasmic extract) was added to 1 mL Trizol for cytoplasmic RNA extraction. The pellet was washed twice with PBS and then treated with Trizol for nuclear RNA extraction. The isolated RNAs were used for cDNA synthesis and qRT-PCR analysis. The primers are listed in Appendix A.

### 2.15. Fluorescence in Situ Hybridization (FISH) Assays

The FISH probes that targeted USP2-AS1, U6, and 18S were purchased from Ribo-biotech, and the FISH assays were performed according to the manufacturer’s instructions.

### 2.16. Bioinformatic Analysis

HNSCC patient RNA-seq data were downloaded from the TCGA data portal. Single-gene GSEA was performed as previously described [21].

### 2.17. Statistical Analysis

Student’s *t*-test was used to analyze differences between groups. Results are presented as means ± SD. Statistical significance was determined as *p* < 0.05. All tests were performed in at least three independent experiments.

## 3. Results

### 3.1. USP2-AS1 Is a Hypoxia-Regulated lncRNA and Predicts Poor Prognosis in HNSCC

To identify the hypoxia-regulated lncRNAs and investigate their roles during HNSCC progression, we first performed RNA-seq using the HNSCC cell line FaDu. Cells were cultured under normoxia (21% O_2_) and hypoxia (1% O_2_) for 12 h, and then total RNAs were collected for RNA-seq (Figure 1A). The RNA-seq results showed that there were 305 lncRNAs upregulated (fold change ≥ 2, FDR ≤ 0.05) and 90 lncRNAs downregulated under hypoxia (Figure 1A and Dataset 1). Among these upregulated lncRNAs, we noticed that HIF1A-AS1 [22], LUCAT [23], UPK1A-AS1 [24], and H19 [25] (Figure 1B) have been reported to possibly be regulated by hypoxia, so these lncRNAs could be positive controls for hypoxia treatment.

To pick out the most functional lncRNAs from these hypoxia-regulated lncRNAs, we analyzed their expression and clinical significance using the TCGA-HNSCC dataset [26]. We found that USP2-AS1 was remarkably upregulated in HNSCC compared to matched normal tissues (Figure 1C), and more than half of HNSCC patients demonstrated a high expression of USP2-AS1 (Figure 1D). Moreover, the combined analysis of the GTEx and TCGA-HNSCC data showed that USP2-AS1 was also significantly upregulated in HNSCC (Figure 1E). In addition, patients with higher USP2-AS1 expression suffered a poorer prognosis (Figure 1F). These results suggested that USP2-AS1 might be a hypoxia-regulated oncogenic lncRNA during HNSCC progression. Previously, USP2-AS1 has been reported as a direct target of MYC [27]. In this study, we knocked down MYC in the HNSCC cell line FaDu and found that USP2-AS1 levels were also decreased upon MYC depletion (Appendix A).

We then examined the subcellular distribution of USP2-AS1 in FaDu cells. Both the qRT-PCR and FISH assays revealed that USP2-AS1 was largely localized in the nuclei of the cells (Figure 1G,H).

### 3.2. USP2-AS1 Is Directly Regulated by HIF1α under Hypoxia

To investigate the molecular mechanism behind the regulation of USP2-AS1 under hypoxia, we first validated the RNA-seq results. As shown in Figure 2A, the expression of USP2-AS1 was elevated by more than five-fold under hypoxia in CAL27 cells, with HIF1A-AS1, H19, LUCAT, PDK1 et al. as positive controls. As HIF1α is the most pivotal transcription factor under hypoxia, USP2-AS1 might be regulated by HIF1α. Next, we knocked down HIF1α using two shRNAs (Figure 2B) in FaDu cells and then treated cells under hypoxia for 12 h. The protein levels of HIF1α were significantly inhibited by both shRNAs (Figure 2B). The qRT-PCR results showed that the depletion of HIF1α remarkably inhibited the elevation of USP2-AS1 under hypoxia (Figure 2C). These results suggested that USP2-AS1 was regulated by HIF1α. Through the analysis of the promoter region (−3000 nt to + 174 nt) of USP2-AS1, we found 11 putative hypoxia-response elements (HRE, A/GCGTG) (Figure 2D). These findings suggested to us that USP2-AS1 might be a direct target of HIF1α. We then performed dual-luciferase reporter assays, as illustrated in Figure 2D. The USP2-AS1 promoter reporters pHREs1+2, and especially the pHREs2 region, could be significantly responsive to hypoxia in FaDu cells (Figure 2D); we also noticed that pHREs1 responded moderately to hypoxia, and it seemed that HIF1α was mainly bound to the pHREs2 region. We further performed ChIP-qPCR assays using FaDu cells. The HIF1α antibodies effectively immunoprecipitated HIF1β (Figure 2E) under hypoxia, and the qPCR analysis of the immunoprecipitated genomic DNA revealed that the USP2-AS1 HRE6-11 promoter region could be enriched by HIF1α antibodies (Figure 2F). These results suggested that USP2-AS1 is directly regulated by HIF1α in HNSCC cells.

### 3.3. USP2-AS1 Promotes HNSCC Progression

To investigate the biological role of USP-AS1, we first performed single-gene gene-set enrichment analysis (sgGSEA) [21] using TCGA-HNSCC RNA-seq data. The sgGSEA revealed that USP2-AS1 was positively correlated with MYC targets, glycosis, hypoxia, and glycosis signaling (Appendix A). These findings suggested that USP2-AS1 might exert its oncogenic effects in the abovementioned aspects during HNSCC progression.

We knocked down USP2-AS1 in FaDu cells using three shRNAs (sh#1/2/3), and qRT-PCR revealed that sh#2 was most effective (Figure 3A). Cell proliferation analysis revealed that the silencing of USP2-AS1 inhibited FaDu cell proliferation under both normoxia and hypoxia (Figure 3B). Cell colony formation assays in normoxia revealed that the depletion of USP2-AS1 significantly inhibited FaDu cell proliferation and growth under normoxia (Appendix A). Cell cycle analysis found that the knockdown of USP2-AS1 resulted in cell-cycle arrest in the G0/G1 stage (Appendix A). Moreover, subcutaneous tumor formation assays showed that the silencing of USP2-AS1 significantly inhibited tumor growth in vivo (Figure 2C,D).

To further confirm these results, we overexpressed USP2-AS1 in CAL27 (Figure 2E). As expected, USP2-AS1 overexpression remarkably increased the proliferation and growth of CAL27 cells (Appendix A), accelerated cell transitions from the G0/G1 stage to the S stage (Appendix A), and enhanced tumor growth in vivo (Figure 3F,G).

Additionally, matrigel-coated trans-well assays showed that USP2-AS1 knockdown inhibited FaDu cell invasion under hypoxia (Appendix A), while USP2-AS1 overexpression promoted CAL27 cell invasion (Appendix A).

To validate the sgGSEA results, we determined the expression of cell proliferation-, death-, and glycosis-associated genes in these subcutaneous tumors and found that the knockdown of USP2-AS1 inhibited the expression of cell-cycle-progression- and glycosis-associated genes (Figure 3H). In contrast, the overexpression of USP2-AS1 enhanced the levels of these mRNAs (Figure 3I). All these results indicated that USP2-AS1 promotes HNSCC progression.

### 3.4. USP2-AS1 Binds to DCAF13 in HNSCC Cells

To understand how USP2-AS1 promotes HNSCC progression, we sought to identify its binding partner in hypoxia-treated FaDu cells. In vitro transcribed biotin-labeled USP2-AS1 and control (anti-sense of USP2-AS1) were used for RNA-pulldown experiments [28] (Figure 4A), and we then identified the differential binding proteins by LC–MS/MS. Most of the potential binding proteins of USP2-AS1 are listed in Figure 4B. Next, we analyzed the clinical significance of these proteins in the TCGA-HNSCC dataset and found that DCAF13 (DDB1- and CUL4-associated factor 13), a substrate receptor for the CUL4–DDB1 E3 ubiquitin–protein ligase complex [29], was also highly expressed (Figure 4C) and positively correlated with USP2-AS1 (Figure 4D): a higher level of DCAF13 predicted poorer prognosis (Figure 4E) in HNSCC patients. Furthermore, DCAF13 was distributed in both the nucleus and cytoplasm of FaDu cells (Appendix A), similarly to USP2-AS1 (Figure 1H,I). These findings allowed us to identity the interaction between DCAF13 and USP2-AS1.

We then validated the binding of DCAF13 and USP2-AS1 in FaDu cells, as shown in Figure 4F. USP2-AS1 pulled down DCAF13, but not the negative control β-actin. In contrast, RIP assays showed that USP2-AS1 could be immunoprecipitated by Flag-DCAF13, but not 18S RNA or U1 snRNA (Figure 4G,H). These findings demonstrated that DCAF13 binds to USP2-AS1 in HNSCC cells. As we know, the function of lncRNA is dependent on its secondary structure [30]. By using the *RNAfold webserver*, we predicted USP2-AS1’s secondary structure and constructed two truncations (T1 and T2) (Figure 4I). Compared to full-length USP2-AS1, both the T1 and T2 truncations, especially T2, abrogated the interaction with DCAF13 (Figure 4I).

### 3.5. USP2-AS1 through DCAF13 Promotes HNSCC Progression

DCAF13 has been reported to function as an oncogenic E3 ligase in breast cancer and osteosarcoma by the targeting and degradation of tumor suppressors PERP [31] and PTEN [32], respectively. In the current study, we used UbiBrowser 2.0, a useful tool for predicting ubiquitin ligase/deubiquitinase–substrate interactions [33], to predict the putative substrates of DCAF13. Unexpectedly, tumor suppressive proteins including P53, CDKN1C, and ATR were listed (Figure 5A and Dataset 2); ATR was also shown to bind to USP2-AS1 according to the RNA-pulldown and LC–MS/MS data (Figure 4B). We then assumed that USP2-AS1 might promote DCAF13 by targeting these putative tumor-suppressive substrates, especially ATR.

We identified the protein level of P53 and ATR under hypoxia in FaDu and CAL27 stably transfected cells. DCAF13 knockdown increased the level of ATR in FaDu cells, whereas DCAF13 overexpression decreased the ATR levels in CAL27 cells (Appendix A). However, the P53 levels did not change in either cell line. Furthermore, we also found that silencing USP2-AS1 elevated the protein level of ATR and P53 under hypoxia but not normoxia, and the results are presented in Appendix A. Moreover, we demonstrated that exogenous Flag-DCAF13 immunoprecipitated both ATR and P53 (Figure 5B). The overexpression of USP2-AS1 inhibited the protein levels of ATR and P53, and DCAF13 depletion restored the protein levels of ATR FaDu cells under hypoxia (Figure 5C). We noticed that elevated levels of USP2-AS1 inhibited the accumulation of P53 under hypoxia.

In addition, the CCK8 cell viability assays showed that the knockdown of DCAF13 restored the cell proliferation and growth by USP2-AS1 overexpression (Figure 5D). As expected, the proliferation of CAL27 was not promoted by the overexpression of the T1 and T2 truncations, which did not bind to DCAF13. These results demonstrated that USP2-AS1 promoted cell proliferation through DCAF13 in HNSCC.

## 4. Discussion

Herein, we reported that lncRNA USP2-AS1 is upregulated under hypoxia and is a direct target of HIF1α. Both loss- and gain-of-function assays revealed that USP2-AS1 promoted cell proliferation and invasion in HNSCC. Moreover, patients with higher levels of USP2-AS1 were associated with undesirable clinical outcomes. These results suggested that USP2-AS1 exerts its oncogenic effects on HNSCC tumorigenesis and progression.

It is becoming increasingly evident that lncRNA deregulation contributes to tumorigenesis and progression [34]. Hypoxia-regulated lncRNAs function as oncogenes in a wide range of cancers, including HNSCC [35]. For example, lncRNA HIFCAR/MIR31HG promotes HNSCC progression by modulating the transcriptional activity of HIF1α [36,37]. In addition, Xiang et al. reported 14 hypoxia-regulated lncRNA signatures that could be used for prognosis prediction [38]. USP2-AS1 has been reported to be possibly regulated by MYC [27]. As a global transcription factor, MYC regulates the expression of a wide range of genes [39]. It has been reported that MYC helps cells survive under low-oxygen conditions, and emerging evidence suggests that MYC and HIF also cooperate to promote cancer cell growth and progression [40]. The regulation of USP2-AS1 by MYC and HIF1α suggests that it might take part in the crosstalk between the MYC and HIF1α signaling pathways during cancer initiation and progression.

lncRNAs regulate gene expression through transcription, post-transcription, translation, and post-translation mechanisms [41,42]. Herein, we demonstrated that USP2-AS1 exerts its oncogenic effect by binding to DCAF13, a substrate receptor for the CUL4–DDB1 E3 ubiquitin–protein ligase complex [29]. The mechanism of lncRNA binding to the E3 ligase and regulating protein stability has been uncovered previously. lncRNA CF129 binds to P53, the E3 ligase, and MKRN1 and then promotes the degradation of P53 [43]. lncRNA SNHG11 binds to VHL and blocks the interaction between VHL and HIF-1α, preventing its ubiquitination and degradation [44]. In the current study, we therefore assumed that USP2-AS1 might regulate DCAF13 by targeting its substrates. We predicted the potential substrates of DCAF13 and identified the tumor-suppressive proteins TP53, CDKN1C, and ATR. We demonstrated that P53 and ATR bind to DCAF13, and DCAF13 regulates the protein level of ATR but not P53. Moreover, the overexpression of USP2-AS1 inhibited and the silencing of USP2-AS1 elevated the protein level of ATR and P53 under hypoxia in CAL27 and FaDu cells, respectively. All these findings suggested that USP2-AS1 might regulate DCAF13 by targeting these tumor suppressors. Nevertheless, the mechanism through which USP2-AS1 affects DCAF13 by targeting its substrates should be further explored. Although the overexpression of USP2-AS1 has previously been associated with colorectal cancer and lung cancer progression [27,45], the current study added another level of complexity to the manner in which USP2-AS1 regulates cancer progression.

lncRNA has potential advantages for cancer diagnosis and prognosis [46,47]. Whether USP2-AS1 could function as an independent risk factor for HNSCC patients should be further investigated. A higher level of USP2-AS1 was strongly associated with the poorer overall survival of HNSCC patients in this study. Considering this evidence, USP2-AS1 has remarkable potential for HNSCC prognosis. Exploring the role of USP2-AS1 in HNSCC and its clinical significance has greatly enhanced our understanding of the molecular pathology of HNSCC and provided a potential treatment target for HNSCC.

## Figures and Tables

**Figure 1 cells-11-03407-f001:**
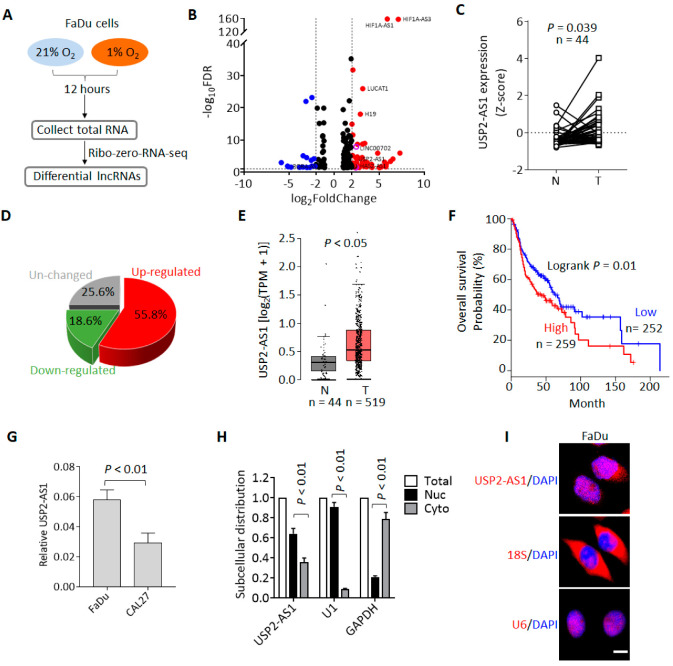
Hypoxia-regulated lncRNA USP2-AS1 is highly expressed in HNSCC. (**A**) Flow chart of RNA-seq for hypoxia and normoxia treatments. (**B**) Volcano plot analysis of the RNA-seq data: red and blue plots indicate hypoxia up- and downregulated lncRNAs (fold change ≥ 1.5, FDR ≤ 0.05), respectively. (**C**) USP2-AS1 in TCGA-HNSCC normal and tumor paired samples: N, normal; T, tumor. (**D**) Pie chart depicting the change in USP2-AS1 in paired HNSCC patients. (**E**) USP2-AS1 in TCGA-HNSCC and GTEx normal and TCGA-HNSCC tumor samples: N, normal; T, tumor. (**F**) Highly expressed USP2-AS1 predicts poorer prognosis in HNSCC patients, data from TCGA-HNSCC dataset. (**G**) qRT-PCR analysis of USP2-AS1 in indicated HNSCC cell lines. (**H**) qRT-PCR analysis of the subcellular distribution of USP2-AS1 in FaDu cells: U1 as nuclear marker, GAPDH as cytoplasm marker. (**I**) FISH assay analysis of the subcellular distribution of USP2-AS1 in FaDu cells: U6 as nuclear marker, 18S RNA as cytoplasm marker. Scale bar, 20 μm; data in (**C**,**G**,**H**) are presented as the mean ± SD of three independent experiments, with *p* value indicated.

**Figure 2 cells-11-03407-f002:**
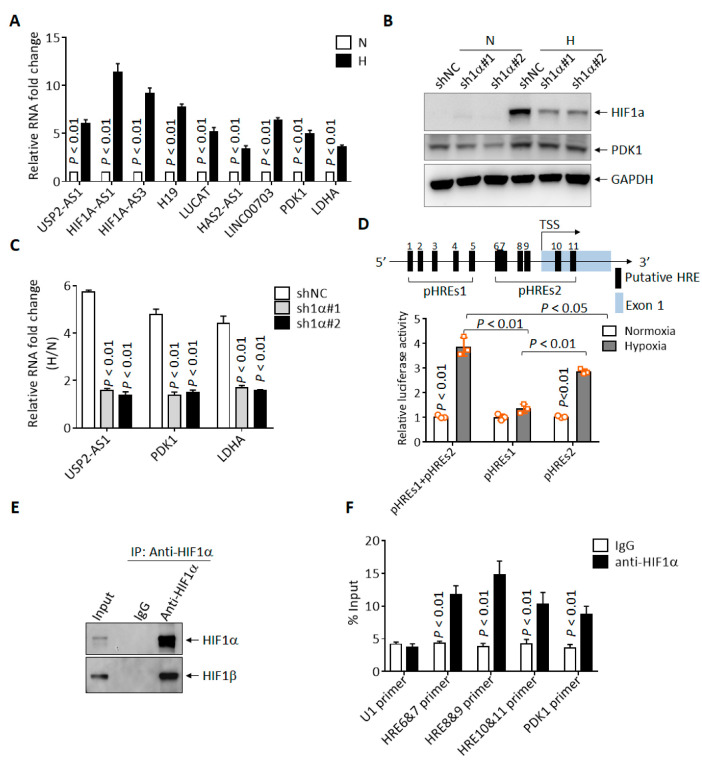
USP2-AS1 is a direct target of HIF1α. (**A**) qRT-PCR analysis of indicated lncRNAs in FaDu cells under normoxia and hypoxia. (**B**) Immunoblotting analysis of HIF1α in FaDu cells with or without HIF1α knockdown: PDK1 as HIF1α-target control. (**C**) qRT-PCR analysis the level of USP2-AS1 in FaDu cells with or without HIF1α knockdown: PDK1 and LDHA as HIF1α-target controls. (**D**) Schematic diagram of USP2-AS1 promoter with hypoxia-responsive elements (HREs) indicated (top): dual-luciferase reporter assays revealed that both pHREs1+2 and pHREs2 USP2-AS1 promoter constructs but not pHRESs1 were responsive to hypoxia (bottom) in HEK293T cells. (**E**,**F**) ChIP-WB analysis revealed that HIF1α antibodies could immunoprecipitate HIF1β (**E**), and ChIP-qPCR analysis revealed that HIF1α bound to indicated USP2-AS1 HRE regions (**F**): U1 as negative control, PDK1 as positive control. Data in (**A**,**C**,**D**,**F**) represent the mean ± SD of three independent experiments, with *p* value indicated.

**Figure 3 cells-11-03407-f003:**
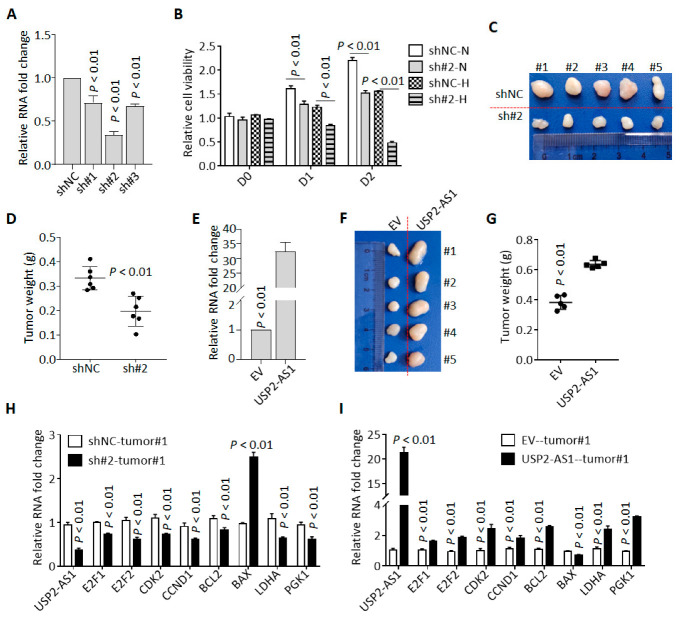
Knockdown of USP2-AS1 inhibits cell proliferation both in vitro and in vivo. (**A**) qRT-PCR analysis of the level of USP2-AS1 in shRNA-transfected FaDu cells: NC, negative control. (**B**) Knockdown of USP2-AS1 dramatically inhibited FaDu cell proliferation under both normoxia (N) and hypoxia (H), as measured by CCK8 cell viability assays. (**C**,**D**) Knockdown of USP2-AS1 inhibited subcutaneous tumor formation by FaDu cells: tumor size (C) and weight (D) are shown. (**E**) qRT-PCR analysis of the level of USP2-AS1 in CAL27 cells with or without USP2-AS1 overexpression: EV, empty vector. (**F**,**G**) Overexpression of USP2-AS1 promoted subcutaneous tumor formation by CAL27 cells: tumor size (F) and weight (G) are shown. (**H**,**I**) qRT-PCR analysis of the cell-cycle-, apoptosis-, and glycosis-related genes in indicated FaDu (H) and CAL27 (I) xenograft tumors. Data in (**A**,**B**,**E**,**H**,**I**) represent the mean ± SD of three independent experiments, with *p* value indicated.

**Figure 4 cells-11-03407-f004:**
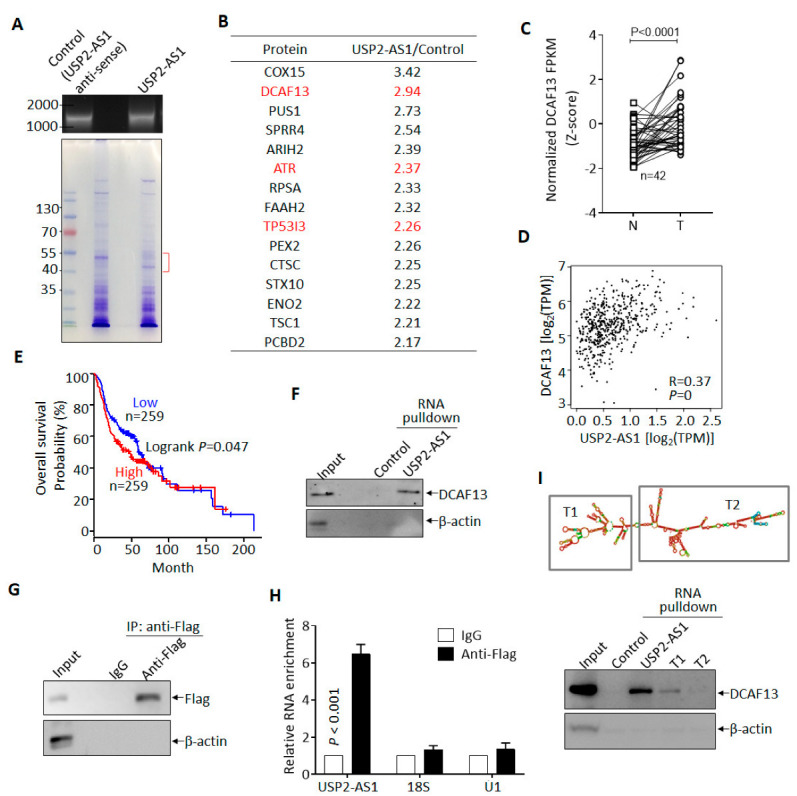
USP2-AS1 binds to DCAF13 in HNSCC cells. (**A**) In vitro transcribed biotin-labeled USP2-AS1 and control (top) were used for RNA-pulldown assays: Coomassie brilliant blue staining of pulled-down samples are shown (bottom). (**B**) LC–MS/MS-identified USP2-AS1-bound proteins are listed. (**C**) DCAF13 in TCGA-HNSCC normal and tumor paired samples: N, normal; T, tumor. (**D**) DCAF13 is positively correlated with USP2-AS1 in HNSCC patients, data from TCGA. (**E**) Kaplan–Meier plot revealed that the high expression of DCAF13 predicts poor prognosis in HNSCC patients, data from TCGA-HNSCC dataset. (**F**) RNA-pulldown revealed that USP2-AS1 binds to DCAF13, but not β-actin. (**G**,**H**) RNA-IP-WB revealed that Flag antibodies immunoprecipitated Flag-DCAF13 (**G**), and RNA-IP-qPCR indicated that Flag-DCAF13 bound to USP2-AS1, but not 18S RNA and U1 snRNA (**H**). (**H**) The secondary structure of USP2-AS1 was predicted using the *RNAfold* website: rectangles indicate the truncated constructs T1 and T2 (top). RNA-pulldown assays revealed that T1 and T2 lost some (T1) or all (T2) of their ability to bind to DCAF13. Data in (**C**,**G**,**H**,**I**) represent the mean ± SD of triplicate experiments.

**Figure 5 cells-11-03407-f005:**
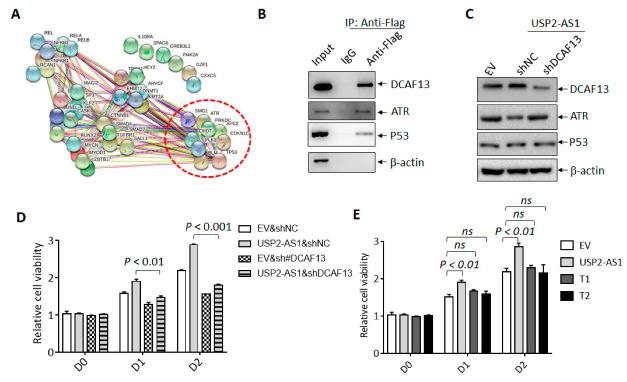
USP2-AS1 through DCAF13 promotes HNSCC progression. (**A**) String analysis of the putative substrates of DCAF13 predicted by UbiBrowser 2.0. (**B**) Flag-co-IP-WB revealed that indicated proteins bound to DCAF13 in FaDu cells under hypoxia for 12 h. (**C**) Immunoblotting analysis of indicated proteins in CAL27 cells with indicated transfection. (**D**,**E**) CCK8 assays analyzed the cell viability of CAL27 cells with USP2-AS1 (**D**), T1, and T2 overexpression under normoxia (**E**). Data in (**D**,**E**) are presented as mean ± SD of triplicate experiments. The experiments in (**B**,**C**) were conducted in triplicate.

## Data Availability

Not applicable.

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
