# Peer review of "Hypoxia-Regulated lncRNA USP2-AS1 Drives Head and Neck Squamous Cell Carcinoma Progression"

_cells, 2022, doi:10.3390/cells11213407_

Round 1
Reviewer 1 Report
Tang et al. investigated the role of the lncRNA USP2 antisense RNA 1 (USP2-AS1) in head and neck squamous cell carcinoma (HNSC). The lncRNA USP2-AS1 was uncovered after screening hypoxia-regulated lncRNA in the HNSC cell line FaDu by RNA-seq. USP2-AS1 expression is transcriptionally regulated by oxygen-dependent transcriptional activator HIF1alpha. Functional assays revealed that USP2-AS1 promotes cell proliferation and invasion in vitro and tumor growth in vivo. Mechanically, RNA-pulldown assays revealed that USP2-AS1 binds to the E3 ligase DCAF13, thus regulating putative tumor suppressive substrates, especially ATR.
Overall, the authors provide novel evidence regarding the relevance of the oncogenic lncRNA USP2-AS1 in HNSC. However, to fully establish their conclusions several issues should be addressed appropriately, as described below:
Major comments
1. Provide control and target-specific shRNA sequences. Target-specific shRNA sequences for MYC and HIF1alpha should also be included.
2. Indicate the concentration of primary antibodies for Western blot assays.
3. Several methods were not described in material and methods. All material and methods shouls be listed.
4. Figure 2D should be plotted as relative activity and not as ratio. Comment about the activity of pHREs1 elements. How luciferase activity was normalized? Include a full description of the experiment in material and methods.
5. Figure 4A requires improvement. Pull-down conditions are unclear. What is highlighted in red?
6. The authors concluded that USP2-AS1 exert its oncogenic role by downregulating the tumor suppressors ATR and p53 involving the E3 ligate complex CUL4-DDB1. The evidence is incomplete, I encourage the authors to test ATR and p53 expression under normoxia and hypoxia in control and USP2-AS1 knock-down cells.
Minor comments
1. I would like to suggest to short the title to “Hypoxia regulated lncRNA USP2-AS1 drives head and neck squamous cell carcinoma progression”
2. Re-write “Cell lines” section. HEK-293T is not a HNSC model.
3. Line 73. Lentivirus transduce cells.
4. Line 353. Improper abbreviation JW-X et al.
Reviewer 2 Report
Head and neck squamous cell carcinoma (HNSCC) comprise a heterogeneous group of tumors that derived from the mucosal epithelium in the oral cavity, pharynx and larynx. Pathogenesis of HNSCC is generally correlated with exposure to tabacco-derived carcinogens and excessive alcohol consumption. In addition, tumors that arise in the oropharynx are attributed to infection with HPV, primarily HPV-16. Genetics factors also contribute to HNSCC risk, however molecular pathogenesis is they are complex and not completely understood. Epigenetic mechanisms (operating among others by lncRNAs) that translate the external influences into specific genes expression provide a pivotal link between environment and genetics and therefore the lncRNAs deregulation might account for HNSCC. This paper present a complex analysis of hypoxia regulated lncRNAs in HNSCC. RNA-seq screen of hypoxia regulated lncRNA in HNSCC revealed up regulation of USP2 antisense RNA1. Further functional analysis indicated USP2-AS1 as HIF1α regulated oncogenic lncRNA that promotes cell proliferation and invasion in vitro and in vivo. In addition RNA-pulldown and mass spectrometry pinpointed E3 ligase DDB1 and CUL4-assoiacted 17 factor as binding partners for USP2-AS1 in HNSC cells.
Major and Minor points that should be addressed to current manuscript:
· Abbreviation of head and neck squamous cell carcinoma is HNSCC
· Manuscript requires linguistic proofreading. Authors should pay attention on word repetition, for example: abstract line 17 “that that”
· What the authors understood by “mechanical analysis”? (line 56)
· In the subsection “Cell lines” the information regarding cells culture under hypoxia should be included
· Authors often used “HNSC cells”, which is very general term. May author define more precisely biologic material, that was investigative in this study?
· Line 134 not “weighed” but weighted
· Line 160 “dismal prognosis” might be replaced by “poor prognosis”
· Figure 1, G and H, is USP2-AS1 expression statistically significant different in study samples?
· Figure 2, B, why for one of the shNC, protein product of HIF1alfa is not observed?
· Figure 2, D, figure description is unclear: “Dual-luciferase reporter assays revealed that indicated USP2-AS1 promoter truncated pGL3.1 229 constructs were responded to hypoxia (bottom)”. May author explain it more precisely.
· Figure 4, E: There is no information regarding HNSC patients groups collected in current study (number of study group, age, sex, diagnosis criteria etc.). What biologic material was examined in the context of gene expression?
Round 2
Reviewer 1 Report
I have not further comments to the authors.
Reviewer 2 Report
There are not other comments and suggestions for author.